# Effects of Storage Time and Thawing Method on Selected Nutrients in Whole Fish for Zoo Animal Nutrition

**DOI:** 10.3390/ani12202847

**Published:** 2022-10-19

**Authors:** Angela Gimmel, Katrin Baumgartner, Sandra Bäckert, Anja Tschudin, Barbara Lang, Anna Hein, Sandra Marcordes, Fabia Wyss, Christian Wenker, Annette Liesegang

**Affiliations:** 1Institute of Animal Nutrition and Dietetics, Vetsuisse Faculty, University of Zurich, 8057 Zurich, Switzerland; 2Zoo Nuremberg, 90480 Nürnberg, Germany; 3Granovit AG, 4303 Kaiseraugst, Switzerland; 4Zoologischer Stadtgarten Karlsruhe, 76137 Karlsruhe, Germany; 5Zoo Duisburg, 47058 Duisburg, Germany; 6Zoo Basel, 4054 Basel, Switzerland

**Keywords:** whole fish analysis, herring (*Clupeus harengus*), mackerel (*Scomber scombrus*), capelin (*Mallotus villosus*), retinol, cholecalciferol, tocopherol, thiamin

## Abstract

**Simple Summary:**

Fish-eating animals in human care receive whole fish that were frozen, stored and thawed before feeding. Nutrient losses have been documented, but exact changes are unknown. The aims of this study were to evaluate whether or not frozen fish lose vitamins and trace minerals during storage, and if different thawing methods have an influence on the degree of these losses. Whole herring, mackerel, and capelin were analyzed at four time points within a storage period of six months at −20 °C. Each time, three thawing methods were tested: refrigerator, room temperature and running water. The following nutrients were analyzed: vitamin A, B1, D3 and E, iron, copper, zinc, and selenium. Copper was below detection limits in all samples, vitamin B1 in most herring (44/48) and capelin samples (25/36), and vitamin D3 in half of the capelin samples (18/36). Significant decreases of vitamin A, D3 and E concentrations were observed during a storage period of six months. Thawing fish with different methods resulted in a significant change of concentration of vitamin A. It is essential to supplement vitamin B1 and E in diets containing whole fish, and it should not be stored longer than 6 months, due to depletion of vitamin A, D3 and E.

**Abstract:**

Piscivores in human care receive whole fish that were frozen, stored and thawed before feeding. Nutrient losses have been documented, but exact changes during storage and with different thawing methods are unknown. Primarily, it was hypothesized that frozen fish lose different vitamins and trace minerals during a storage period of six months. Secondly, that different thawing methods have a significant influence on the degree of vitamin loss. Three fish species, herring (Clupeus harengus), mackerel (Scomber scombrus) and capelin (Mallotus villosus) were analyzed at four time points within a storage period of 6 months at −20 °C. At each time point, three thawing methods were applied: thawing in a refrigerator (R), thawing at room temperature (RT), and thawing under running water (RW). The following nutrients were analyzed: vitamin A, B1, D3 and E, iron (Fe), copper (Cu), zinc (Zn) and selenium (Se). The statistical method used was a linear mixed effect model. Cu was below detection limits in all analyzed samples, vitamin B1 in most analyzed herring (44/48 samples) and capelin (in 25/36 samples), respectively. In addition, the vitamin D3 concentration was also below detection limits in half of the capelin samples (18/36). No concentration changes of Fe (*p* = 0.616), Zn (*p* = 0.686) or Se (*p* = 0.148) were observed during a storage period of six months, in contrast to a significant decrease in vitamin A (*p* = 0.019), D3 (*p* = 0.034) and E (*p* = 0.003) concentrations. Thawing fish with different thawing methods did not result in concentration changes of Fe (*p* = 0.821), Zn (*p* = 0.549) or Se (*p* = 0.633), but in a significant concentration change of vitamin A (*p* = 0.002). It is essential to supplement vitamins B1 and E in diets containing whole fish to avoid deficiencies in piscivorous species, and care should be taken not to store fish longer than six months, due to the depletion of vitamins A, D3 and E.

## 1. Introduction

A whole-fish diet is a staple for various animal species in human care, spanning all classes of vertebrates and including cartilaginous fishes, bony fishes, amphibians, reptiles, birds, and mammals [1]. All these species can be summarized as piscivores, which refers to the feeding behavior of consuming fish, including, in a broader sense, a variety of fresh- and saltwater invertebrates [1]. Fresh whole prey represents a complete and balanced diet for these piscivores, and meets their nutritional requirements [1]. In human care, the majority of piscivorous species receive whole fish that were caught, frozen, shipped to the next storing facility, stored until ordered, shipped to the buyer, stored again and finally thawed just before feeding [2]. Therefore, the nutritional status of piscivores in human care who rely entirely on a diet of selected species of whole fish is not only dependent on the quality and nutrient composition of the fish that was caught, but also on the process of freezing, shipping, storing, and thawing that follows [2]. The freezing process leads to microorganisms being prevented from continuing their multiplication. This however cannot be re-evaluated when buying fish from different fish companies. Nutrient depletion during the handling and storage of fish has been documented in numerous case reports of nutritional deficiencies in piscivores in human care [3]. In the chapter Nutrition and Energetics of the *CRC Handbook of Marine Mammal Medicine*, three out of five major nutritional disorders cover vitamin deficiencies or toxicities [4]. This study examined the two last steps in the whole-fish production process (storing at a zoological garden and thawing before feeding), since the zoo staff can directly influence these steps; the examination of these phases can offer valuable information to ameliorate nutritional deficiencies associated with fish storage and fish handling.

As early as 1955, vitamin B1 deficiency was reported in a grey seal (*Halichoerus grypus*) [5] and California sea lions (*Zalophus californianus*) [6]. Piscivores are especially susceptible to vitamin B1 deficiency, since it is broken down by the enzyme thiaminase, which is naturally present in different parts of many fish species, such as herring and smelt [4]. Vitamin B1 deficiency is a threat to the health of marine mammals in human care, as illustrated in a recent report of mortality in harbor seals fed fish with high levels of thiaminase [7].

The fat-soluble vitamins A, D and E are thought to be abundant in marine organisms [3]. Whereas the vitamin A level in whole fish probably meets the vitamin A requirements of piscivores, analyses of vitamin E levels and numerous reports of vitamin E deficiency in piscivorous species in human care demonstrate that vitamin E levels in stored and thawed whole fish are likely to be inadequate [3]. This is due to the antioxidative nature of vitamin E, which depletes while counteracting peroxidation in whole fish high in polyunsaturated fatty acids. [4]. Since a high vitamin A supplementation can interfere with the absorption of vitamin D and vitamin E [4], the following vitamins were assessed in this study: vitamin A, B1, D3 and E. 

Not much information can be found on trace minerals in whole fish. Therefore, this study examined the minerals iron (Fe), copper (Cu), zinc (Zn) and selenium (Se). Selenium acts as a cellular antioxidant, and was of special interest, as it counteracts peroxidation in whole stored fish, and also has a sparing effect on vitamin E, hence its association with antioxidant depletion over time in storage [8]. 

Gimmel et al. showed in 2016 that most European facilities caring for bottlenose dolphins (*Tursiops truncatus*) store their fish for a period of three to six months (10/19 facilities) and use different thawing methods, including thawing in a refrigerator (R) (6/19 facilities), thawing at room temperature (RT) (3/19 facilities), and thawing under running water (RW) (1/19 facilities), as well as a combination of above-mentioned methods (9/19 facilities) [9]. The main fish species used were herring (*Clupeus harengus)* (16/19 facilities), capelin (*Mallotus villosus*) (16/19 facilities), and squid (*Ilex and Loligo sp.*) (16/19 facilities). Since the focus of this study lay exclusively on fish species, mackerel *(Scomber scombrus*) was included instead of squid. Therefore, the fish species herring (*Clupeus harengus)*, capelin (*Mallotus villosus*) and mackerel *(Scomber scombrus*), a storage period of six months and the three thawing methods mentioned above, were examined in this study. 

The following two hypotheses were developed: firstly, it was hypothesized that frozen fish lose vitamins and trace minerals that function as antioxidants during a storage period of six months. Secondly, that the thawing method has an influence on the vitamin concentration of whole fish, and water-soluble vitamins are reduced when running water is used to thaw whole fish. The objective and aim of this study was to give recommendations on valuable supplementation as well as practical thawing procedures. 

## 2. Methods

Four different zoological gardens participated in this study (Z1, Z2, Z3 and Z4). Each institution ordered whole fish from their usual supplier, and the fish were delivered in 20–25 kg blocks of frozen whole fish. All zoological gardens ordered herring (*Clupeus harengus*), mackerel (*Scomber scombrus*) and capelin (*Mallotus villosus*) with the exception of Z4, which only ordered herring (*Clupeus harengus*) and mackerel (*Scomber scombrus*). Each block of frozen whole fish consisted of one catch of fish. On delivery, one block of frozen fish per species was set aside for this study, and stored at the zoological garden at −20 °C. The age of the fish on delivery ranged from 26–297 days after catch. Nutritional analyses were conducted at four different time points during storage: delivery (T1), after 60 days (T2), 120 days (T3) and 180 days (T4). The tested thawing methods were: thawing in a refrigerator (R, 4 °C for 24 h), thawing at room temperature (RT, 18–22 °C for 15 h overnight) and thawing under running water (RW, 14° C water temperature for 3 h with the water running). At T1 to T4, 1 kg of whole fish per fish species and thawing method was separated from the frozen block, either by chain saw or by pickaxe, thereby keeping the fish frozen. If a chain saw was used, care was taken that the block that was cut out contained at least 1 kg of whole frozen fish. If a pickaxe was used, care was taken not to damage the whole fish. The still-frozen fish was then sent with an express frozen parcel service to the laboratory (AGROLAB LUFA GmbH, Kiel, Germany), to ensure that it arrived frozen throughout. In the laboratory, each kilogram of whole fish per species was thawed with a different thawing method (R, RT, RW). After thawing, 1 kg of whole fish was homogenized in a blender, and subjected to analysis. The thawing water was discarded and not included in the nutritional analysis. Nutritional analysis (two replicates) for each different thawing method was conducted, and the following parameters were analyzed: moisture content, vitamin A, vitamin B1, vitamin D3, vitamin E as well as the four trace minerals Fe (Fe), zinc (Zn), Cu (Cu), and selenium (Se). 

### 2.1. Nutritional Analysis

Moisture content was analyzed by adding an appropriate quantity of anhydrous sand, according to Commission Regulation EC 152/2009, Annex III, 4.1.3. for feed in liquid or paste form and feed predominantly composed of oils and fats. The moisture content was expressed in percentage of original matter. 

Vitamin A was analyzed by reverse phase high performance liquid chromatography (HPLC) with UV and fluorescence detection, according to Commission Regulation EC 152/2009, Annex IV, and included all-trans-retinyl alcohol and its cis-isomers, which are determined by this method. The concentration of vitamin A was expressed in international units (IU) per kg. One IU corresponds to the activity of 0.300 μg all-trans-vitamin A alcohol, 0.344 μg all-trans-vitamin A acetate or 0.550 μg all-trans-vitamin A palmitate. The limit of quantification was 2000 IU vitamin A/kg original matter. 

Vitamin B1 was analyzed by reverse phase HPLC after post-column derivatization of vitamin B1 to thiochrome and fluorescence detection, according to the method DIN EN 14122:2010 (Deutsches Institut für Normung, Berlin, Germany, 2010). The concentration of vitamin B1 was expressed as mg vitamin B1 hydrochloride per kg whole fish. The limit of quantification was 0.2 mg vitamin B1/kg original matter.

Vitamin D3 was analyzed by reverse phase HPLC with UV detection, according to the method VDLUFA III, 13.8.1, a modification of the standardized method DIN EN 12821 (Deutsches Institut für Normung, 2009). The concentration of vitamin D3 was expressed in international units (IU) per kg. One IU corresponds to the activity of 0.025 μg cholecalciferol. The limit of quantification was 300 IU vitamin D3/kg original matter. 

Vitamin E was analyzed by reverse phase HPLC with UV-and fluorescence detection, according to Commission Regulation EC 152/2009, Annex IV and the concentration of vitamin E was expressed as mg DL-α-tocopherol acetate per kg. One mg DL-α-tocopherol acetate corresponds to 0.91 mg DL-α-tocopherol. The limit of quantification was 2 mg vitamin E/kg original matter.

Fe and Zn were analyzed by inductively coupled plasma optical emission spectrometry (ICP-OES), according to the method DIN EN 15621 (Deutsches Institut für Normung, 2010), and the concentration of each trace element was expressed as mg/kg. 

Cu and Se were analyzed by inductively coupled plasma mass spectrometry (ICP-MS), according to the method DIN EN ISO 17053, and the concentration was expressed as mg/kg. The limit of quantification was 1 mg/kg original matter for Cu.

To allow for comparison between samples, the nutritional analyses results were converted to dry matter basis (DMB).

### 2.2. Statistical Analysis

For statistical analysis, R version 3.5.0 was used (R Core Team, 2018). To test if the fish species herring (*Clupeus harengus*), mackerel (*Scomber scombrus*) and capelin (*Mallotus villosus*) differ regarding their nutrient concentration, a one-way analysis of variance (ANOVA) was applied, modeling the different nutrients as a function of the different fish species (example: aov (Fe~species)). The dataset used was whole fish nutrient concentrations at T1 using thawing method R. To test if storage time of herring (*Clupeus harengus*), mackerel (*Scomber scombrus*) and capelin (*Mallotus villosus*) has an influence on their nutrient concentration, a linear mixed effect model (LMER) was applied, where the time point and the fish species were included as fixed effects, and the batch of fish (including fish species, zoological gardens and time point) was included as random effect (example: lmer(Fe~species + timepoint + (1|batch))). The dataset used was whole fish nutrients concentration using thawing method R. To test if the thawing method of herring (*Clupeus harengus*), mackerel (*Scomber scombrus*) and capelin (*Mallotus villosus*) has an influence on their nutrient concentration, a linear mixed effect model (LMER) was applied, where the time point, the fish species and the thawing method were included as fixed effects, and the batch of fish (including fish species, zoological garden and time point) was included as random effect (example: lmer(Fe~species+timepoint+thawing+(1|batch))). The complete data set, inclusive of data retrieved from all thawing methods and time points, was used in this model.

## 3. Results

### 3.1. Nutritional Analysis

The Cu concentrations of the examined fish species herring (*Clupeus harengus*), mackerel (*Scomber scombrus*) and capelin (*Mallotus villosus*) were below detection levels (<1.0 mg/kg original matter) in all samples, as was the vitamin B1 concentration in almost all analyzed herring (*Clupeus harengus*) (below <0.2 mg/kg original matter in 44/48 analyses) and most analyzed capelin (*Mallotus villosus*) (in 25/36 analyses). The Fe, Se, vitamin B1, and vitamin A concentrations were highest in mackerel (*Scomber scombrus*), and the Zn and vitamin E concentrations were highest in capelin (*Mallotus villosus*), whereas it contained the lowest vitamin D3 concentrations. In addition, the vitamin D3 concentration was below detection levels (<300 IU/kg original matter) in 18/36 whole capelin (*Mallotus villosus*) analyses. However, none of these differences were statistically significant apart from vitamin E (*p* = 0.0015) and vitamin A (*p* = 0.039).

### 3.2. Influence of Storage Time

No concentration changes of Fe (*p* = 0.616), Zn (*p* = 0.686), Se (*p* = 0.148) or vitamin B1 (*p* = 0.495) were observed during a storage period of six months in whole herring (*Clupeus harengus*), mackerel (*Scomber scombrus*) and capelin (*Mallotus villosus*). However, significant decreases in vitamin A (*p* = 0.019), vitamin D3 (*p* = 0.034) and vitamin E (*p* = 0.003) concentrations were observed during these six months (Figure 1, Figure 2 and Figure 3). The loss of vitamin E was most apparent in capelin (*Mallotus villosus*).

### 3.3. Influence of the Thawing Method

Thawing fish with the different thawing methods (R, RT, RW) did not significantly influence the concentration of Fe (*p* = 0.821), Zn (*p* = 0.549), Se (*p* = 0.633), vitamin D3 (*p* = 0.551), vitamin E (*p* = 0.913) or vitamin B1 (*p* = 0.688) in whole herring (*Clupeus harengus*), mackerel (*Scomber scombrus*) and capelin (*Mallotus villosus*). It could be shown however, that the concentration of vitamin A (*p* = 0.002) changed significantly using different thawing methods (Figure 4). 

## 4. Discussion

This study mainly focused on how selected nutrients, if not administered in appropriate amounts, may lead to either deficiencies or over-supplementation. In the case of thiamine and vitamin E, this may lead to different neurological symptoms, while in the case of vitamins A or D, this may lead to skeletal deformations. In this study, one of the hypotheses tested was whether the storage time of frozen fish has an influence on different micronutrients. The freezing process itself was not investigated, since this would have gone beyond the scope of the study, although this might also influence the micronutrient content at the very start.

An unexpected finding of this study was that the Cu concentration of the examined fish species herring (*Clupeus harengus*), mackerel (*Scomber scombrus*) and capelin (*Mallotus villosus*) was below detection levels, e.g., below 1.0 mg/kg original matter. Bernhard et al., 2002, showed levels of 4–6 mg/kg on DMB for herring (*Clupeus harengus*), 5–10 mg/kg DMB for mackerel (*Scomber scombrus*) and 3–10 mg/kg DMB for capelin (*Mallotus villosus*) [1]. In original matter, these results would equal 1–1.5 mg/kg, 1.6–3.3 mg/kg, and 0.6–2 mg/kg [1]. In short, the published Cu concentrations of the examined fish species should have been detectable in this study. Cu requirements for piscivores are not very well defined, but roughly range from 4 to 10 mg/kg DM diet [10,11,12], which could be met by the published Cu levels of whole fish in Bernhard et al., 2002, but would likely not be met by the Cu concentrations analyzed in this study. The question that arises is the following: should Cu be supplemented if piscivorous species in Europe are fed herring (*Clupeus harengus*), mackerel (*Scomber scombrus*) and capelin (*Mallotus villosus*)? Clinical signs of Cu deficiency include anemia, hair depigmentation, skeletal deformities, aortic rupture, ataxia, infertility, and diarrhea [8]. No reports of Cu deficiency are available for any species of aquatic organisms, whereas reports on Cu toxicity on aquatic organisms are numerous, especially for invertebrates and fish [13]. Other possibilities for undetectable Cu concentrations might firstly be an insufficient analytical method. However, this is unlikely, since ICP-MS is a United States Environmental Protection Agency-recognized technique for environmental trace element analyses, and has been used for Cu analysis in other studies that examine Cu in whole fish [14,15]. The second possibility includes the depletion of Cu between catching the fish at sea and feeding the fish to animals in human care. The fish was not freshly caught when arriving at the zoological garden. It was caught, frozen onboard the fishing vessel, shipped to the next storing facility, and stored until ordered. The age of the fish at delivery ranged from 26–297 days after the catch, a time in which it is possible for Cu depletion to occur. Therefore, further studies are necessary in this area, determining the ideal analytical method and examining Cu depletion between catching and feeding the fish to animals. 

The presence of thiaminase in herring (*Clupeus harengus*) and capelin (*Mallotus villosus*) explains why the vitamin B1 concentration was often below the detection limit of 0.2 mg/kg original matter in these fish species. Herring (*Clupeus harengus*) has been demonstrated to contain thiaminases [4] and they were also present in whole fish, head, skin, muscle, and viscera of capelin (*Mallotus villosus*) [16]. Croft et al., 2013, showed that the activity of thiaminase in whole capelin (*Mallotus villosus*) is lower than in herring (*Clupeus harengus*) (5.1 μg of thiamine consumed/h and g of fish (wet weight) compared with 8.7 μg, respectively) [8]. These results are in line with our study, since more vitamin B1 could be found in capelin (*Mallotus villosus*) than herring (*Clupeus harengus*). 

Significant decreases in the concentration of the tested fat-soluble vitamins (vitamin A, D3 and E) in whole herring (*Clupeus harengus*), mackerel (*Scomber scombrus*) and capelin (*Mallotus villosus*) were observed within a storage period of six months. The loss of vitamin E was most apparent in capelin (*Mallotus villosus*), which was probably because this fish species contained the highest level of vitamin E in the first place in DMB. Vitamin E acts as an antioxidant [1,4,8]. When fat (in the form of long-chain polyunsaturated fatty acids) reacts with oxygen, free fatty radicals are produced [8]. Once produced, these radicals can react with other fatty acids, resulting in a chain reaction. Vitamin E protects the cells from oxidative damage, and thereby depletes [1,4,8]. The longer a fish is stored, the more vitamin E is depleted [1,4,8]. This study confirms the depletion of Vitamin E during a storage period of six months, and therefore, vitamin E should be supplemented. In addition, this study showed that vitamin A and vitamin D3 concentrations also deplete. 

Vitamin A is considered an unstable vitamin and it is sensitive to light, heat, and oxygen [2]. Vitamin A requirements for piscivores are not very well defined but range from 1.1 to 9 IU/g DM diet [10,11,12]. The lowest measured vitamin A concentration of whole fish in this study was 5.2 IU/g DM whole fish. Therefore, although vitamin A concentrations decreased, vitamin A supplementation would not seem necessary if whole herring (*Clupeus harengus*), mackerel (*Scomber scombrus*) and capelin (*Mallotus villosus*) that are stored for 6 months at −20 °C, are fed. Other authors also reached the same conclusion [3,4,9].

Vitamin D3 is also a potent antioxidant that facilitates balanced mitochondrial activities, preventing oxidative stress-related protein oxidation, lipid peroxidation, and DNA damage [17]. Vitamin D3 requirements for piscivores range from 0.25 to 0.75 IU/g DM diet [10,11,12]. The lowest measured vitamin D3 concentration of whole fish in this study was 2.7 IU/g DM whole fish. It appears that, although vitamin D3 concentrations decreased, vitamin D3 supplementation would not seem necessary if whole fish that are stored for 6 months at −20 °C are fed. However, it must be considered that in 18/36 capelin samples, vitamin D3 was below the detection limit of 300 IU/kg original matter. Also, Gimmel et al., 2016, showed that calcidiol blood concentrations of bottlenose dolphins (*Tursiops truncatus*) in human care were lower, compared with values from free-range animals, even though no negative health implications for the tested animals from low serum calcidiol concentrations were detected in that study [9]. Therefore, the authors would recommend the supplementation of vitamin D3 if fish were stored longer than 6 months or if capelin were used as a main dietary source.

In this study, the different microorganism species that regain their activity during the thawing process will not be discussed, due to the fact that they were not further investigated. Thawing fish with the different thawing methods (R, RT, RW) only had a significant influence on the concentration of vitamin A (*p* = 0.002) in whole herring (*Clupeus harengus*), mackerel (*Scomber scombrus*) and capelin (*Mallotus villosus*) in our study. This finding was very unexpected, as fish that was thawed with RT had the highest vitamin A concentration compared with R and RW. Reasons include an increased loss or degeneration of vitamin A during thawing with R or RW, or an increased conversion of vitamin A from a form that was not detected by the chosen laboratory method to a form that was detected. Vitamin A is an unstable vitamin and sensitive to light, heat, and oxygen [2]. Therefore, it seems unlikely that RT would lead to a reduced loss of vitamin A compared with R and RW, as light, heat and oxygen are probably highest with the thawing method RT. The chosen laboratory method measured all-trans-retinyl alcohol and its cis-isomers. However, there are many more forms of vitamin A. Retinoids are a class of compounds that are forms of vitamin A [18]. They can exist in a variety of forms, including retinal (the aldehyde form of vitamin A), retinol (the alcohol form of vitamin A), retinoic acid (the carboxylic form of vitamin A), and retinyl ester (the ester form of vitamin A) [18]. Retinoid can be transformed into an appropriate form (retinal, retinol, retinoic acid, or retinyl ester) in the organ by retinoid-converting enzymes during metabolism, while some conversions are reversible and some are irreversible [18]. The optimal temperatures of retinoid-converting enzymes are in the range of 25–40 °C [18], which would be closest to the temperature for the thawing method RT. These retinoid-converting enzymes can be found in bacteria and in different tissues of an organism [18]. The bacterial contamination of fish is highest with thawing at RT [2]. It could therefore be possible that the thawing method by itself does not influence vitamin A concentration in whole fish per se, but that other mechanisms such as vitamin A form conversion and bacterial contamination are the reason for this. Therefore, further studies are necessary, where the drip water is analyzed for its vitamin A concentration using different thawing methods. For now, the influence of the thawing method on the vitamin A concentration of whole fish is not really of practical relevance, since whole fish probably contains enough vitamin A. Regarding feed hygiene, the thawing method R should be used, as other studies have shown [2,9,19].

No significant influence of thawing method could be shown on vitamin B1 (*p* = 0.688). This was probably due to the small number of fish samples that had a detectable vitamin B1 concentration. More research needs to be conducted in respect of water-soluble vitamins and thawing methods, and care should be taken to choose a fish species without thiaminases, such as mackerel.

## 5. Conclusions

Firstly, a storage period of six months at −20 °C reduced vitamin A, vitamin D and vitamin E concentrations in whole frozen herring (*Clupeus harengus*), mackerel (*Scomber scombrus*) and capelin (*Mallotus villosus*), whereas it did not influence trace mineral concentrations. Secondly, the tested thawing methods had no influence on the vitamin D and vitamin E concentration of whole frozen herring (*Clupeus harengus*), mackerel (*Scomber scombrus*) and capelin (*Mallotus villosus*). The influence on vitamin B1 concentrations could not be evaluated properly, due to an insufficient sample size. The influence on vitamin A concentration was significant, while thawing at room temperature showed the highest vitamin A concentration in the fish. 

For practical purposes, we conclude the following: iq1qt is essential to supplement vitamin E and B1 in diets containing whole fish, especially herring (*Clupeus harengus*) and capelin (*Mallotus villosus*), to avoid deficiencies in piscivorous species. The authors would also recommend the supplementation of vitamin D3 if the fish were stored longer than 6 months, or if capelin were used as a main dietary source. Based on this study, no recommendations can be made with regards to thawing method on selected nutrients in whole fish for zoo animal nutrition.

## Figures and Tables

**Figure 1 animals-12-02847-f001:**
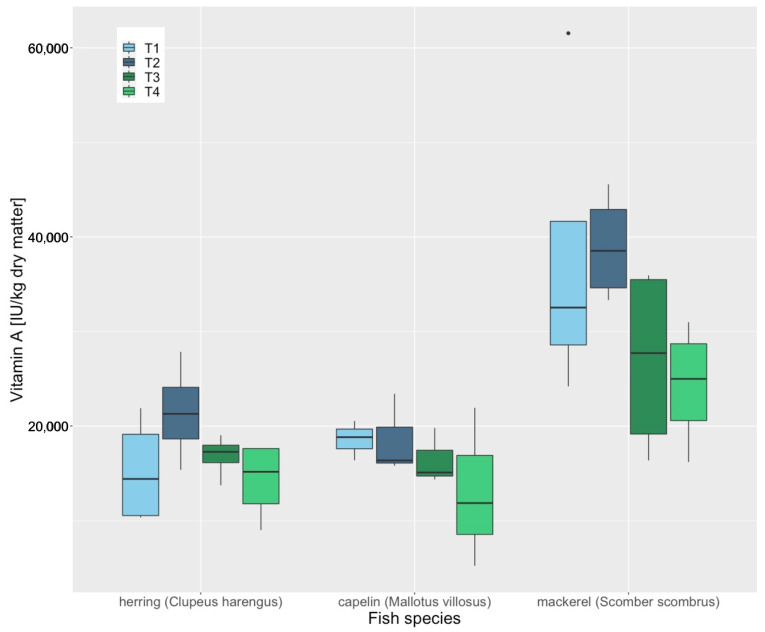
Vitamin A concentrations of whole frozen herring (Clupeus harengus), mackerel (Scomber scombrus) and capelin (Mallotus villosus), thawed in the refrigerator (4 °C for 24 h), and measured at 4 different time points. T1, day of delivery, T2, 60 days after, T3, 120 days after, T4, 180 days after delivery. “•” are outliners.

**Figure 2 animals-12-02847-f002:**
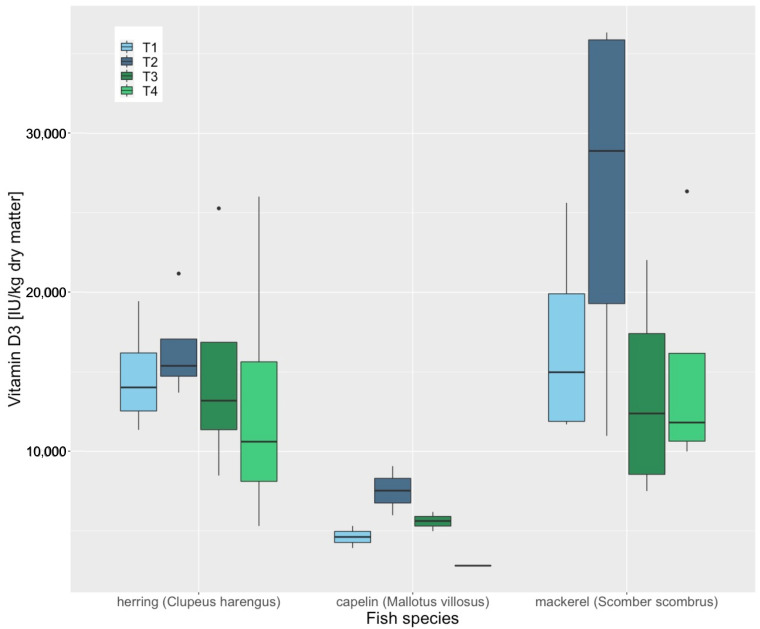
Vitamin D3 concentrations of whole frozen herring (Clupeus harengus), mackerel (Scomber scombrus) and capelin (Mallotus villosus), thawed in the refrigerator (4 °C for 24 h), measured at 4 different time points. T1, day of delivery, T2, 60 days after, T3, 120 days after, T4, 180 days after delivery. “•” are outliners.

**Figure 3 animals-12-02847-f003:**
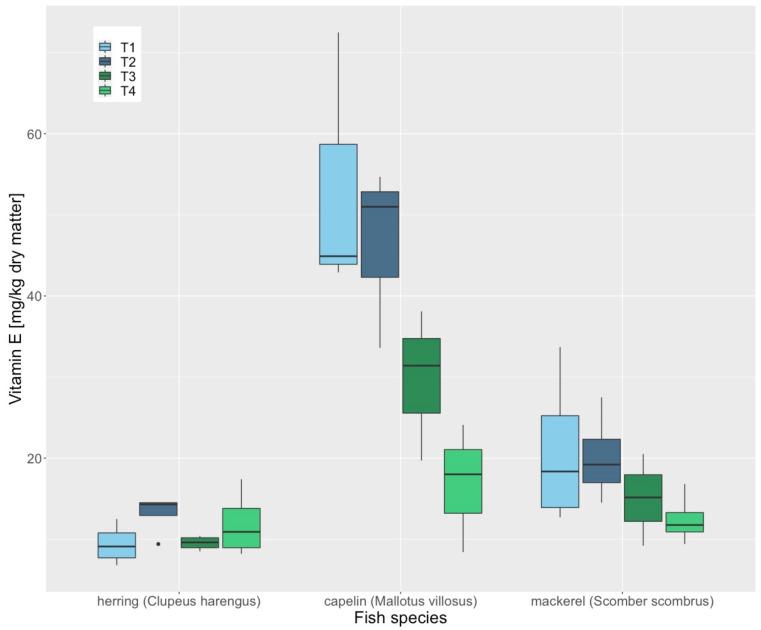
Vitamin E concentrations of whole frozen herring (Clupeus harengus), mackerel (Scomber scombrus) and capelin (Mallotus villosus), thawed in the refrigerator (4 °C for 24 h), measured at 4 different time points. T1 is day of delivery, T2, 60 days after, T3, 120 days after, T4, 180 days after delivery. “•” are outliners.

**Figure 4 animals-12-02847-f004:**
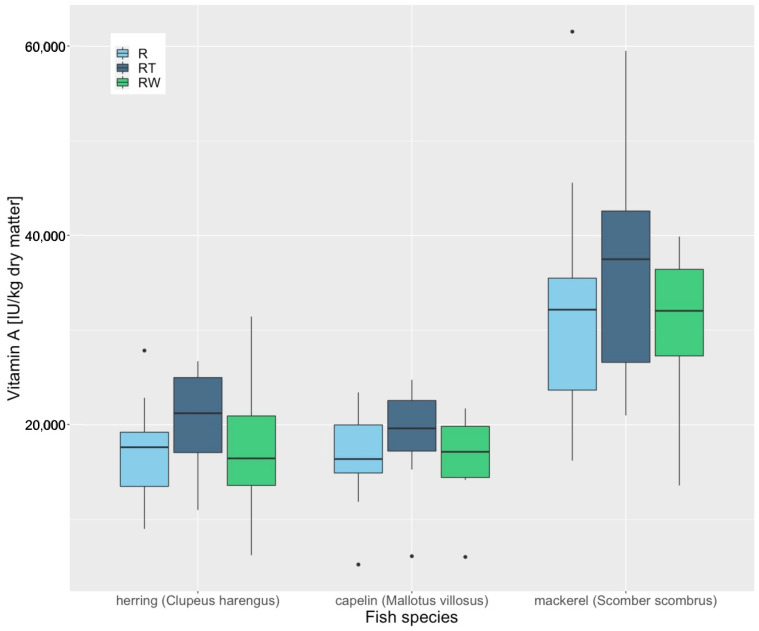
Vitamin A concentrations of whole frozen herring (Clupeus harengus), mackerel (Scomber scombrus) and capelin (Mallotus villosus), thawed with different thawing methods. R = thawing in a refrigerator (4 °C for 24 h), RT = thawing at room temperature (18–22 °C for 15 h overnight) and RW = thawing under running water (14 °C water temperature for 3 h with the water running). “•” are outliners.

## Data Availability

The data presented in this study are available on request from the corresponding author. The data are not publicly available due to confidentiality reasons.

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
