# Peer review of "Effects of Storage Time and Thawing Method on Selected Nutrients in Whole Fish for Zoo Animal Nutrition"

_animals, 2022, doi:10.3390/ani12202847_

Round 1
Reviewer 1 Report
The only spelling mistake I have found: line 241: dot instead of comma (Bernhard et al, )
Beside that, the paper is well written and can be published.
Author Response
Reviewer #1: Responses to comments:
Thank you very much for your statement stating that the paper is well written and can be published.
Comment: Line 241: dot instead of comma (Bernhard et al, )
Response: Thank you for pointing out our mistake. It has been corrected and a period has been included.
Reviewer 2 Report
It is necessary to include in the experimental design the number of samples that were made and how many replicates there were.
which are the ones that already exist and take care of the quality of defrosted products.
It is necessary to discuss the process of freezing food, since it does not cause the loss of its nutrients, as can be seen in the discussions. It has an effect on microorganisms, which are the main responsible for the deterioration of the frozen product.
It is necessary to discuss that in freezing, microorganisms are prevented from continuing their multiplication. It is not mentioned in the discussions that when the fish is thawed, the microorganisms resume their activity.
It is important to carry out studies of the thawing methods specifically for this type of samples, which are used for food in zoos, it is mentioned but there is no mention of alternative thawing methods that have already been studied and the loss of nutrients is avoided .
Author Response
Reviewer #2: Responses to comments:
Thank you for your input to ameliorate this manuscript. We have tried to answer to your suggestion accordingly.
Comment: lt is necessary to include in the experimental design the number of samples that were made and how many replicates there were.
Response: Thank you for your comment – we included the replicates in the material and methods part. The number of samples take were from each zoo, each fish species and each timepoint – please refer to line 114.
Comment: Which are the ones that already exist and take care of the quality of defrosted products.
Response: Thank you for your comment, which is not so clear to us. There is no authority that takes care of defrosted products – the zoo itself is responsible for the hygiene and the quality control. Once in a while the fish are analysed with a proximate analysis.
Comment: It is necessary to discuss the process of freezing food, since it does not cause the loss of its nutrients, as can be seen in the discussions. It has an effect on microorganisms, which are the main responsible for the deterioration of the frozen product.
Response: Thank you for the comment. We now added a sentence that is stating that the freezing process itself may also be problem, depending on how fast this is realized. But that was not the scope of this study. The interest and hypothesis were if the storage time has an influence on the micronutrients. In the discussion, it is mentioned that the low B1 content of some fish already at the first samplepoint might be due to the fact that some fish naturally contain thiaminase I originating not necessarily from bacteria but naturally found in the viscera of some fish. Refer to line 171.
Comment: It is necessary to discuss that in freezing, microorganisms are prevented from continuing their multiplication. It is not mentioned in the discussions that when the fish is thawed, the microorganisms resume their activity.
Response: We now added two sentences – one in the introduction and one in the discussion that when fish is thawed microorganisms regain their activity. We want to state that this was however not the interest of this study and was not examined.
Comment: It is important to carry out studies of the thawing methods specifically for this type of samples, which are used for food in zoos, it is mentioned but there is no mention of alternative thawing methods that have already been studied and the loss of nutrients is avoided .
Response: Thank you very much for this useful comment. Yes, we are indeed aware of other thawing methods for fish, which have been investigated for special fish as mackerel for human consumation. Also the positive effects of these thawing methods. But this was not in the scope of our study, since the thawing in a zoo facility has to be practical and also from the work load (really large amounts of fish has to be thawed at once for a large number of fish eaters) as well the costs has to be makeable. For human fish as food this might really play an important role and is much more important. In fact, the method that is recommended to be the best in humans (low temperature thawing) is also the method that we recommend, although it is time consuming and the keepers have to think of the fish to be put into the low temperature rooms. We have now added another reference.
Reviewer 3 Report
The manuscript is very interesting. The methodology used is adequate. The results support the discussion. However, I have the following comments.
I. major comments:
1. The authors present results on the content and changes of micronutrients. However, it would be very interesting if the authors could refer to macronutrients, especially proteins and fatty acids.
2. In the introduction it would be good if the authors include a brief paragraph on macronutrients.
3. In the discussion, I suggest including a point about the possible nutritional deficiencies that could be generated, and how it would be possible to correct them.
II. Minor comments:
1. Improve the writing of the objective of the study.
2. In the title replace "nutrients" with "micronutrients"
Author Response
Reviewer #3: Responses to comments:
Thank you for the valuable statement at the beginning of your review and inputs which helped us to ameliorate the manuscript.
Comment: The authors present results on the content and changes of micronutrients. However, it would be very interesting if the authors could refer to macronutrients, especially proteins and fatty acids.
Response: The focus of our study was really the micronutrients and how these change during storage time as well depending on the different thawing methods. Especially in fish eaters in zoos this is always an issue since a lot of unuseful supplements are given that may even lead to an overnutrition (if you think of Vitamin A e.g.). So we wanted to follow our hypothesis.
Comment: In the introduction it would be good if the authors include a brief paragraph on macronutrients.
Response: Macronutrients were already focused on in different studies. This would have gone beyond the scope of our study as we had a focus on the problems that rose from practical zoo nutrition.
Comment: In the discussion, I suggest including a point about the possible nutritional deficiencies that could be generated, and how it would be possible to correct them.
Response: Thank you for this valuable content. We included 2 sentences that include the possible nutritional deficiencies.
Comment: Improve the writing of the objective of the study
Response: We tried to improce the objectives and aims of this study and hope that they are now according your expectations.
Comment: In the title replace "nutrients" with "micronutrients".
Response: According to the Word Health Organization (WHO), micronutrients are defined as vitamins and minerals needed by the body in very small amounts. Since these amounts are species specific, the authors feel that they cannot prove for all discussed species, that the selected analyzed nutrients fall within the category of micronutrients. Therefore, the authors would prefer to keep “selected nutrients” in the title.
Round 2
Reviewer 2 Report
Complies with the observations
Reviewer 3 Report
Authors answered all my comments. Therefore, manuscript can be accepted in the present form.